

# Diet and mitochondrial DNA haplotype of a sperm whale (*Physeter macrocephalus*) found dead off Jurong Island, Singapore

Marcus A.H. Chua[1,2], David J.W. Lane[1], Seng Keat Ooi[3], Serene H.X. Tay[4] and Tsunemi Kubodera[5]

[1] Lee Kong Chian Natural History Museum, National University of Singapore, Singapore, Singapore
[2] Department of Environmental Science and Policy, George Mason University, Fairfax, VA, United States of America
[3] Tropical Marine Science Institute, National University of Singapore, Singapore, Singapore
[4] Civil and Environmental Engineering, National University of Singapore, Singapore, Singapore
[5] National Museum of Nature and Science, Tokyo, Japan

Corresponding author
Marcus A.H. Chua,
marcus.chua.ah@gmail.com

## ABSTRACT

Despite numerous studies across the large geographic range of the sperm whale (*Physeter macrocephalus*), little is known about the diet and mitochondrial DNA haplotypes of this strongly female philopatric species in waters off Southeast Asia. A female sperm whale found dead in Singapore waters provided the opportunity to study her diet and mitochondrial DNA haplotype. Here we report on the identification of stomach contents and mitochondrial DNA haplotype of this individual, and we include coastal hydrodynamic modelling to determine the possible geographic origin of the whale. At least 28 species of prey were eaten by this adult female whale, most of which were cephalopods. The mesopelagic squids *Taonius pavo, Histioteuthis pacifica, Chiroteuthis imperator,* and *Ancistrocheirus lesueurii* made up over 65% of the whale's stomach contents. Plastic debris was also found in the whale's stomach. Based on the diet, genetics, and coastal hydrodynamic modelling that suggest an easterly drift of the whale carcass over several days, the dead sperm whale in Singapore probably originated from a pod in the Southern Indian Ocean. This study provides an increase in the understanding the diet and natural history of the sperm whale in Southeast Asia. The combined analyses of stomach contents, DNA, and hydrodynamic modeling could provide a context to future studies on the sperm whale strandings, and have broader applicability for other marine mammals in the region.

## INTRODUCTION

The sperm whale *Physeter macrocephalus* Linnaeus, 1758 has a large geographic range encompassing all oceans in temperate and tropical waters (*Rice, 1989*). They are typically found in regions with deep, off-shore waters, and their diet, which consists mainly of cephalopods—with regional differences in species composition—has been studied in the Atlantic, central, eastern and northern Pacific, and Southern Oceans (e.g.,

*Clarke, 1956*; *Okutani et al., 1976*; *Okutani & Satake, 1978*; *Kawakami, 1980*; *Clarke & Young, 1998*; *Best, 1999*; *Smith & Whitehead, 2000*; *Evans & Hindell, 2004*; *Garibaldi & Podesta, 2014*; *Harvey, Friend & McHuron, 2014*). Pelagic and benthic fishes (actinopterygii and chondrichthyes), crustaceans, and tunicates are also eaten in smaller quantities, indicating different modes of foraging (*Kawakami, 1980*; *Best, 1999*; *Evans & Hindell, 2004*; *Santos et al., 2002*). Sperm whale population structure and genetic studies of the maternally-inherited mitochondrial DNA have shown differentiation between populations, suggesting female philopatry (*Richard et al., 1996*; *Lyrholm et al., 1999*; *Engelhaupt et al., 2009*; *Mesnick et al., 2011*; *Alexander et al., 2016*).

However, even though sperm whales are present in waters off Southeast Asia, which is surrounded by the Indian Ocean and western Pacific Ocean (Indo-West Pacific), the diet and mitochondrial DNA haplotypes of sperm whales there are largely unknown to science. Live sperm whales have been recorded off northwest Malay Peninsula, the Lesser Sunda archipelago, the Sunda Strait of the Indian Ocean, and the South China Sea north of Borneo (*Anderson & Kinze, 1999*; *Barnes, 2005*; *De Boer, 2000*) (Fig. 1). Strandings and carcasses are known from Borneo, Java, the Lesser Sunda archipelago, northwest of the Malay Peninsula, Papua, Raja Ampat Islands, Sulawesi, and Sumatra (*Anderson & Kinze, 1999*; *Anon, 2017*; *Anon, 2018*; *Cahya & Simanjuntak, 2017*; *Mustika et al., 2009*; *Mueanhawong, 2012*; *Ponnampalam, 2012*). Despite their widespread distribution and numerous physical records in the Indo-West Pacific, little else is known about sperm whales in the region.

On 10 Jul 2015, a 10.6 m long adult female sperm whale with erupted mandibular and maxillary teeth (size and tooth eruption suggesting sexual maturity) was found dead off the coast of Jurong Island (Fig. 1), Singapore, which provided an opportunity to salvage the carcass, and conduct a scientific study on aspects of the natural history of the whale. The objectives of this study were to (1) determine the diet of the sperm whale from stomach contents by morphological identification of remains, (2) describe the relative proportion of each prey species and the size of cephalopods eaten by the whale, (3) determine the mitochondrial DNA control region haplotype of this sperm whale, and (4) investigate the possible location of the whale's origin using a coastal hydrodynamic model. This study provides the first description of the diet and mitochondrial DNA haplotype of a sperm whale from Southeast Asia, could provide a context for future studies of sperm whale strandings, and have broader applicability for other marine mammals in the region.

## MATERIALS & METHODS

### Stomach contents collection, morphological identification and analysis

Stomach contents were collected from an adult non-pregnant female sperm whale of 10.6 m body length found dead off the coast of Jurong Island, Singapore (1°16′48.23″N 103°43′57.23″E) on 10 July 2015 (Fig. 1). The subject was identified as an adult based on the body size and presence of erupted mandibular and maxillary teeth. The carcass was towed to a section of a beach not accessible to the public and lifted to land by a lorry crane for defleshing and skeleton preservation. Approval to work on the section of the beach

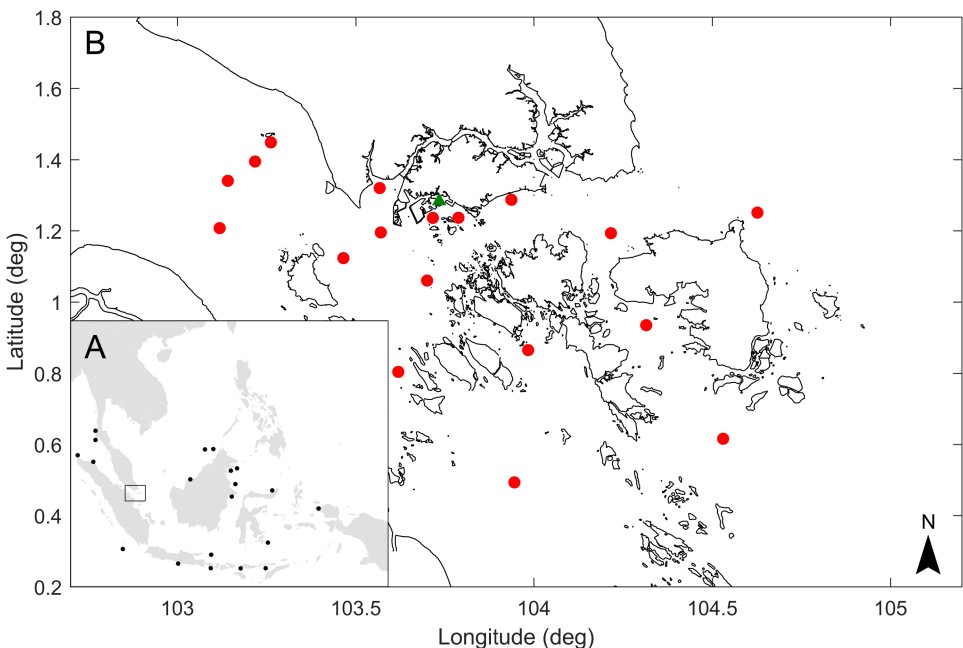

**Figure 1** **Map of sperm whale records in waters off Southeast Asia, location where the sperm whale was found in Singapore and approximate release points of simulated drogues representing the floating dead whale.** (A) Black dots represent sperm whale records in Southeast Asia (*Anderson & Kinze, 1999*; *Barnes, 2005*; *De Boer, 2000*; *Anon, 2017*; *Anon, 2018*; *Cahya & Simanjuntak, 2017*; *Mustika et al., 2009*; *Mueanhawong, 2012*; *Ponnampalam, 2012*). (B) Green triangle shows the location where the sperm whale carcass was found in Singapore, and red dots show the approximate release points in the vicinity of Singapore of simulated drogues.

was granted verbally by Boon Kiang Chew from the National Environment Agency of Singapore. Examination of the whale revealed a copiously bleeding large gash on the back and associated spinal injury, with a few of the caudal vertebrae smashed, possibly caused by a ship strike. Owing to the rapid deterioration of the carcass in tropical conditions after removal from seawater, a standard necrosis was not possible.

Approximately 80% of the total stomach contents were collected. The stomach contents were washed and preserved in 70% ethanol. Pre-sorting of the stomach contents, in particular, upper and lower cephalopod beaks was first performed, and identification of lower cephalopod beaks was later determined by one of the co-authors (TK), and following *Kubodera (2005)*, and *Xavier & Cherel (2009)*. Other diet items were sorted and identified to the lowest taxonomic level by the authors (MAHC and DJWL), and biologists from the Lee Kong Chian Natural History Museum, National University of Singapore.

The average estimated dorsal mantle length (DML) and mass of individuals of each cephalopod species were calculated from the lower rostral length of beaks where conversion formulas were available (*Kubodera, 2005*; *Xavier & Cherel, 2009*; *Clarke, 1986*; *Rodhouse et al., 1990*; *Lu & Ickeringill, 2002*). For species represented by more than 100 beaks, a sample average of 100 beaks was taken. Diet items were counted, and expressed as a percentage total (PT) by number of all prey items present.

## Mitochondrial DNA haplotype identification

Samples of the sperm whale skin and skeletal muscle were collected and frozen at −20 °C. Samples of DNA were extracted from each tissue type using QuickExtract (Epicentre, Madison, WI, USA) following the manufacturer's protocols. Two replicate for each tissue type was done to minimize the likelihood of reporting erroneous sequences because of sequencing error. A section of the mitochondrial DNA control region was targeted using primers and PCR protocols from *Southern, Southern & Dizon (1988)* with a negative control. The resulting PCR product was visualized under UV light after GelRed$^{TM}$ agarose gel electrophoresis. Successful product from PCR was purified using SureClean (Bioline). Cycle sequencing was performed using BigDye Terminator PCR (Applied Biosystems, Foster City, CA, USA) in both directions following the manufacturer's instructions. The resulting single-stranded DNA were purified with CleanSEQ magnetic beads (Agencourt Bioscience Corp), and sequenced on an ABI 3100xl genetic analysis sequencer (Applied Biosystems). The resulting sequences were aligned and edited using the software Sequencher (Gene Codes Corporation, Ann Arbor, WI, USA), and haplotype matching followed *Engelhaupt et al. (2009)*.

## Coastal hydrodynamic modelling

Calibrated coastal hydrodynamic models are useful tools for investigating flow circulation in complex coastal environments. The flow circulation of the coastal waters surrounding the Singapore region are relatively complex due to the impact of tidal mixing, seasonal monsoons and larger tropical storm or depression systems (*Sin et al., 2016*). To be able to capture this, a relatively large domain model with a fine resolution grid of 2 km in the region of interest was used. This South China Sea Model was built in the Delft3D Modelling Framework and was used as the hydrodynamic model for this study as it provides a good representation of tidal and seasonal forcing in the Singapore Strait and the surrounding region (*Tay et al., 2016*). The model is particularly capable of simulating distinct seasonal throughflows in the straits of Singapore and Malacca which was required for the purposes of this study.

To serve as a proxy for a floating dead whale, inert particles called drogues were released during the model simulation. The drogues were released in the model over a seven day period which is assumed to be the maximum flotation time of the dead whale. This was based on the condition of the whale at the time of carcass discovery (Code 2 or early Code 3) (*Geraci & Lounsbury, 1993*), which would suggest a floating time of a week or less in tropical conditions. To examine the possible location where the sperm whale became deceased, drogues were released in various locations in the model seven days prior to its discovery off Jurong Island (Fig. 1). The pathway of drogues that ended up close to Jurong Island on the landfall date were identified as the possible pathways of the floating dead whale.

# RESULTS

## Diet analysis

Morphological sorting and identification revealed 1,835 upper beaks and 1,657 lower beaks of at least 25 cephalopod species (11 identified to species), forming the bulk of

 

the stomach contents (Table 1; Table S1). All diet remains were highly digested, with no fresh tissue. Squids (order Teuthida) formed over 97% of the percentage total by number. *Taonius pavo* was the species with the highest percentage total (31.4%), and at an average estimated mass of 1.11 kg per individual, was probably the most important prey item, followed by *Histioteuthis pacifica* (19.1%; estimated weight not available), *Chiroteuthis imperator* (8.34%; 323 g), and *Ancistrocheirus lesueurii* (7.07%; 325 g). Together these species comprised over 65% of the whale's diet remains numerically. The range of average dorsal mantle length (DML) and mass for the species consumed, estimated with beak conversion formulas, was 11.5–63.9 cm, and 60.4–5,360 g.

The stomach contents included *Pyrosomatidae* material (Tunicata: Thaliacea) comprising nine intact, cylindrical specimens of *Pyrosoma atlanticum* Péron, 1804 (Fig. 2) with colony lengths (post-mortem/ethanol preserved) of 29–100 mm (mean = 66.6 mm). Even though the zooids had been digested, and attempts to obtain 18S DNA for GenBank (NCBI, NIH) matching were unsuccessful, there is a high degree of confidence in identification as the opaque colony tunic had resisted digestion, and its characteristics closely matched the description by van Soest (*Van Soest, 1981*) for *P. atlanticum* (i.e., colony size and shape; zooids densely packed and irregularly arranged; distinct blunt test processes—Fig. 2).

Other than cephalopods and tunicates, an unidentified Thalassinidea (decapod crustacean) cheliped, and unidentified Teleostei (fish) bones were also recorded among the stomach contents.

Non-food items, namely plastic debris were also found in the sperm whale stomach (Fig. 3A). These included plastic drinking cups, food wrappers, and a plastic bag. Two of these items appeared to be of Indonesian origin (Figs. 3B, 3C).

### Mitochondrial DNA haplotype identification

Only the skin samples yielded DNA that could be successfully amplified by PCR. The resulting sequences of a pair of replicates were identical, and fully matched Haplotype A (GenBank accession number DQ512921.1) in *Engelhaupt et al. (2009)*.

### Coastal hydrodynamic modelling

The results of the drogue tracks from the hydrodynamic model simulations are shown in Fig. 4 and the Supplemental Information 1. Figure 4A shows the tracks or paths of all the drogues released seven days prior to the dead whale being discovered. The results generally agree with the expectation that for the particular time of year the predominant currents are eastward through the Singapore Strait. Figure 4B shows the release points and the tracks of a select set of drogues released 7 days prior to the date of the carcass discovery. Given the discovery of the whale off Jurong Island, the whale was likely to have been free floating in the region bordered by the purple, red and blue drogue west of Singapore.

## DISCUSSION

Results from the study support the understanding that the sperm whale is a predator mainly of small to medium-sized squids, with a smaller proportion of other marine invertebrates

**Table 1  Number, percentage total by number (PT), average estimated dorsal mantle length (DML) and estimated mass of prey items found in the sperm whale stomach.**

| Species | Number | PT (%) | Ave. est. DML (cm) | Ave. est. mass (g) | Reference |
|---|---|---|---|---|---|
| Mollusca | | | | | |
| Ancistrocheiridae | | | | | |
| *Ancistrocheirus lesueurii* | 118 | 7.08 | 17.2 | 325 | *Clarke (1986)* |
| Chiroteuthidae | | | | | |
| *Asperoteuthis acanthoderma* | 25 | 1.50 | N.A. | N.A. | |
| *Chiroteuthis imperator* | 139 | 8.34 | 23.8 | 323 | *Clarke (1986)* |
| *Chiroteuthis* sp. A | 115 | 6.90 | 13.4 | 60.4 | *Clarke (1986)* |
| Cranchidae | | | | | |
| *Megalocranchia maxima* | 1 | 0.06 | N.A. | N.A. | |
| *Taonius* cf. *belone* | 15 | 0.900 | 42.5 | 142 | *Rodhouse et al. (1990)* |
| *Taonius pavo* | 523 | 31.4 | 54.9 | 1,110 | *Rodhouse et al. (1990)* |
| *Taonius* sp. A | 3 | 0.18 | 53.4 | 238 | *Rodhouse et al. (1990)* |
| Histioteuthidae | | | | | |
| *Histioteuthis inermis* | 115 | 6.90 | 12.3 | N.A. | *Lu & Ickeringill (2002)* |
| *Histioteuthis pacifica* | 318 | 19.1 | 11.5 | N.A. | *Lu & Ickeringill (2002)* |
| *Histioteuthis* sp. A | 45 | 2.70 | 17.9 | N.A. | *Lu & Ickeringill (2002)* |
| Octopeuthidae | | | | | |
| *Taningia danae* | 6 | 0.36 | 63.9 | 5,360 | *Clarke (1986)* |
| Unidentified Octopeuthidae | 115 | 6.90 | N.A. | N.A. | |
| Onychoteuthidae | | | | | |
| *Onykia loennbergii* | 31 | 1.86 | N.A. | N.A. | |
| Pholidoteuthidae | | | | | |
| *Pholidoteuthis massyae* | 17 | 1.02 | 26.9 | 479 | *Clarke (1986)* |
| Unidentified Teuthida | | | | | |
| Unidentified A | 3 | 0.180 | N.A. | N.A. | |
| Unidentified B | 2 | 0.120 | N.A. | N.A. | |
| Unidentified C | 6 | 0.360 | N.A. | N.A. | |
| Unidentified D | 3 | 0.180 | N.A. | N.A. | |
| Unidentified E | 2 | 0.120 | N.A. | N.A. | |
| Unidentified F | 2 | 0.120 | N.A. | N.A. | |
| Unidentified G | 1 | 0.0600 | N.A. | N.A. | |
| Unidentified H | 3 | 0.180 | N.A. | N.A. | |
| Unidentified I | 9 | 0.540 | N.A. | N.A. | |
| Alloposidae | | | | | |
| *Haliphron atlanticus* | 40 | 2.40 | N.A. | 425 | *Xavier & Cherel (2009)* |
| Arthropoda | | | | | |
| Decapoda | | | | | |
| Unidentified Thalassinidea | 1 | 0.0600 | N.A. | N.A. | |
| Chordata | | | | | |
| Pyrosomatidae | | | | | |
| *Pyrosoma atlanticum* | 9 | 0.540 | N.A. | N.A. | |

**Table 1** (*continued*)

| Species | Number | PT (%) | Ave. est. DML (cm) | Ave. est. mass (g) | Reference |
|---|---|---|---|---|---|
| Actinopterygii | | | | | |
|    Unidentified Teleostei | N.A. | N.A. | N.A. | N.A. | |

**Notes.**
   N.A., not available.

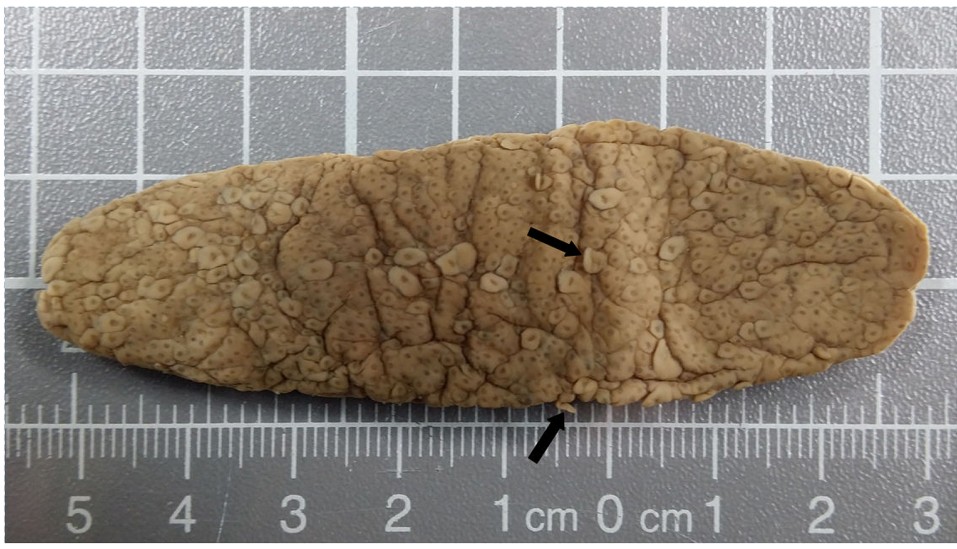

**Figure 2** *Pyrosoma atlanticum* **from stomach of deceased sperm whale.** Colony 85 mm × 25 mm in size. The open end of the somewhat flattened, opaque colony is to the left. Some of the protruding zooid test processes (arrows) are clearly visible.

and fish. The number of species of prey, and relative importance of cephalopods found in the stomach of the Singapore whale is similar to that of sperm whales in the northeastern and southeastern Atlantic, and southwestern Pacific (*Best, 1999*; *Smith & Whitehead, 2000*; *Clarke & Young, 1998*). However, it differed from male sperm whales off Iceland with a high representation of fish eaten (*Martin & Clarke, 1986*), and the whales from seas partially enclosed by landforms (e.g., Mediterranean Sea), which typically have less than 10 prey species recovered (*Garibaldi & Podesta, 2014*; *Santos et al., 2002*; *Roberts, 2003*). This could reflect the availability of prey in the waters where the different sexes of sperm whales forage.

The majority of cephalopod prey species eaten by the sperm whale prior to death are distributed mesopelagically (200–1,000 m) in the Indo-West Pacific (e.g., *Asperoteuthis acanthoderma*, *Chiroteuthis imperator*, *Histoteuthis pacifica*), with some having a wider or global distribution (e.g., *Ancistrocheirus lesueurii*, *Haliphron atlanticus*, *Taningia danae*) (*Chun, 1908*; *Voss, Nesis & Rodhouse, 1998*; *Norman, Nabhitabhata & Lu, 2016*). This, together with the relatively high diversity of prey indicate that the whale was foraging outside the relatively shallow waters within the Singapore Strait (mostly <100 m deep) or surrounding enclosed seas (e.g., Java or South China Sea) (*Voris, 2000*; *Tan et al., 2016*).

Most of the cephalopod prey species consumed were small to medium-sized (1–6% of whale's length) squids with bioluminescent organs, consistent with findings of other studies

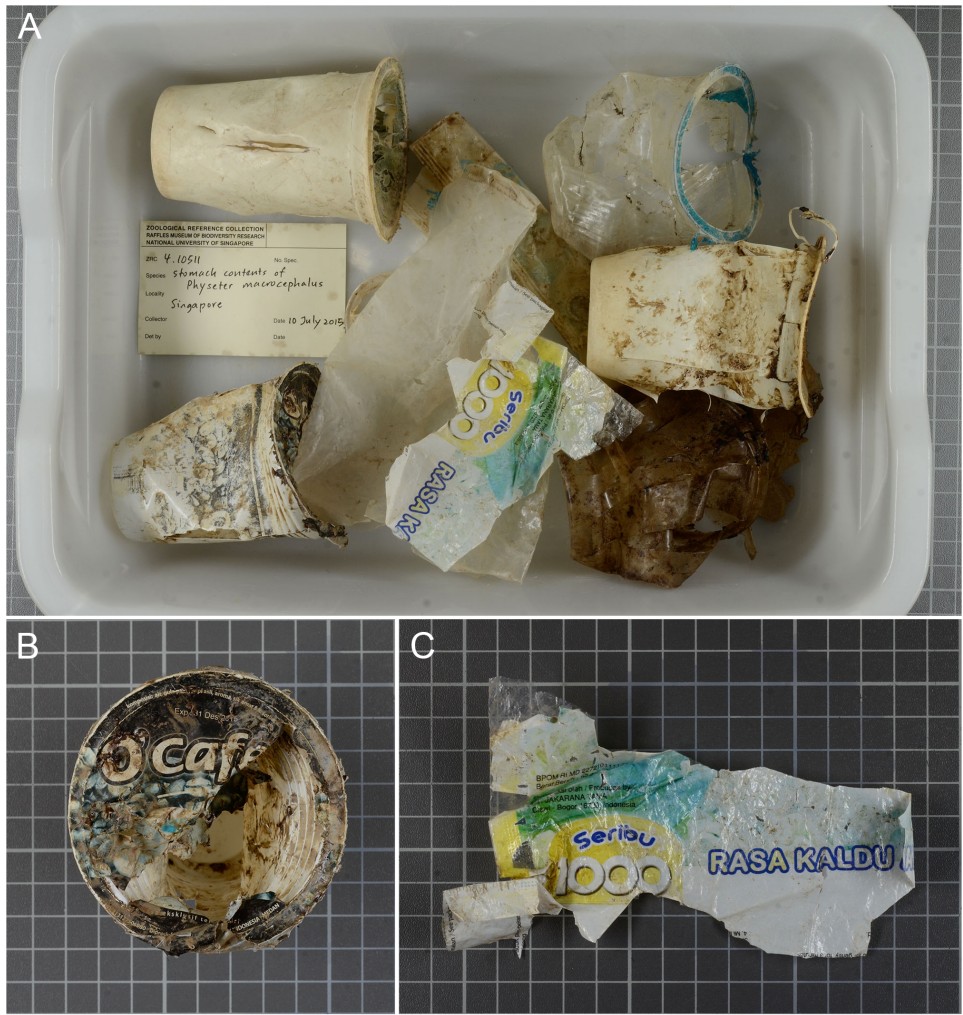

**Figure 3** (A) Plastic debris found in the sperm whale stomach. Scale: each square measures 1 × 1 cm. (B) Drinking cup, and (C) food wrapper with origins from Indonesia.

across oceans (*Kawakami, 1980*; *Clarke & Young, 1998*; *Evans & Hindell, 2004*; *Harvey, Friend & McHuron, 2014*; *Clarke, Martin & Pascoe, 1993*). In this study, bioluminescent photophores are present in a majority of the prey species, i.e., *Ancistrocheirus* species, *Asperoteuthis acanthoderma*, *Chiroteuthis* species, *Histioteuthis* species, *Megalocranchia maxima*, Taningia *danae*, *Taonius pavo* (*Lu, 1977*; *Herring, 1987*; *Herring, Dilly & Cope, 2002*; *Kubodera, Koyama & Mori, 2007*), but absent in *Onykia loennbergii* (*Kubodera, Piatkowski & Okutani, 1998*). Sperm whales are known to forage at depth in the aphotic mesopelagic zone using echolocation (*Watwood et al., 2006*), but with up to 77.5% of cephalopod prey species reported to possess luminous organs, *Clarke, Martin & Pascoe (1993)* suggested it is probable that sperm whales detect and capture most of their food using a combination of echolocation and vision while approaching and swimming through shoals of bioluminescent slow-swimming squids.

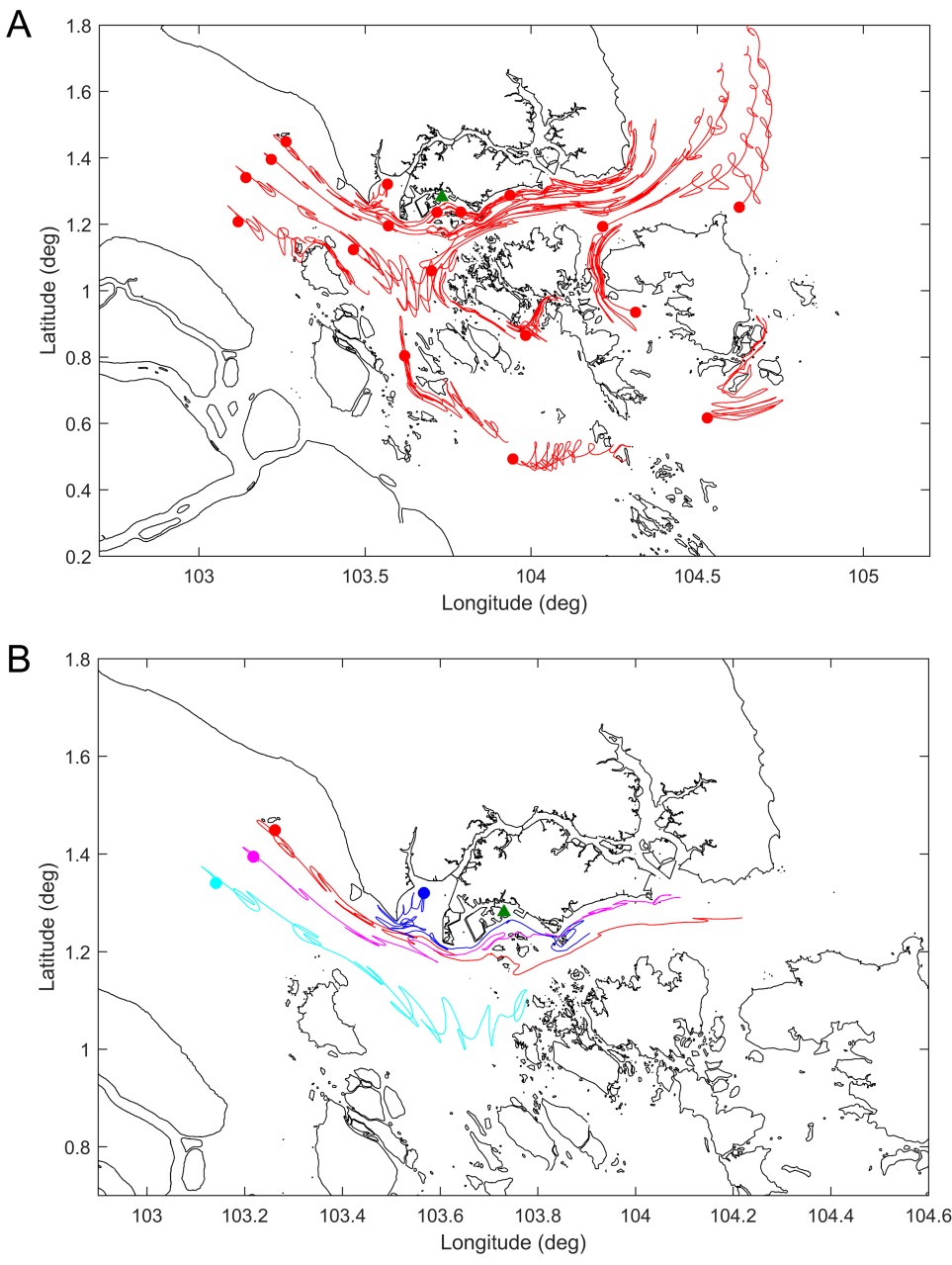

**Figure 4** **Tracks of the simulated drogues over seven days.** (A) All the released drogues; (B) the likeliest drogue tracks to have ended up on the southwestern coast of Singapore on 10 July 2015. The green triangle represents the location where the whale carcass was found. Dots represent the location of the drogues at the start of the simulation.

The hypothesized foraging strategy of sperm whales for bioluminescent squids may also explain the presence of the planktonic, bioluminescent, colonial tunicate, *Pyrosoma atlanticum*, in the diet of this whale. *Pyrosoma atlanticum* has a distribution (50°N–50°S in all oceans) similar to that for female and juvenile sperm whales, as well as mature males

for at least part of the male's life cycle (*Van Soest, 1981*; *Bowlby, Widder & Case, 1990*). *Pyrosoma atlanticum* also occurs over a depth range (0–965 m) comparable to that of squid prey (*Anderson & Sardou, 1994*). These pyrosomes grow to a size of 60 cm by 6 cm (*Van Soest, 1981*) which is within the size range of squid eaten by sperm whales in this and other studies (*Clarke & Young, 1998*; *Evans & Hindell, 2004*; *Harvey, Friend & McHuron, 2014*; *Clarke, Martin & Pascoe, 1993*) and, furthermore, their bioluminescence has been noted to be intense and sustained (*Bowlby, Widder & Case, 1990*; *Wrobel & Mills, 1998*). Thus, strongly bioluminescent *P. atlanticum* colonies occupy the same mesopelagic niche (*Van Soest, 1981*; *Anderson & Sardou, 1994*) as squid prey and it is possible that these tunicates are tracked visually in the same way.

Elsewhere, pyrosomatid colonies have been recorded as prey items in the stomachs of sperm whales captured during whaling operations off the Azores (*Clarke, 1956*; *Clarke, Martin & Pascoe, 1993*) and South Africa (*Best, 1999*). Off South Africa, 73 of 1,268 whales captured (5.76%) contained *Pyrosoma* colonies (*Best, 1999*). Usually the numbers of colonies per whale stomach are small, as found in the present necropsy. Interestingly, diet data from whaling studies indicate that it is exclusively (*Clarke, Martin & Pascoe, 1993*) or predominantly (92%: *Best, 1999*) males that consume these planktonic tunicate colonies, whereas pyrosomatids in this study were consumed by a mature female whale. Sperm whale captures in the South African fishery were typically biased towards males but annual capture inventories were large (>1,000) and of 291 females only two were reported to have consumed *Pyrosoma* (*Best, 1999*). *Best (1999)* considered the consumption of *Pyrosoma* by sperm whales to be opportunistic feeding on a secondary prey item. It is not known whether *P. atlanticum* would be taken in large numbers when these colonial tunicates occur in superabundant swarms (*Wrobel & Mills, 1998*; *Lebrato & Jones, 2009*; *Milstein, 2017*; *Welch, 2017*) but this is a possibility. The importance of pyrosomes, and other pelagic tunicates (*Henschke et al., 2016*) in the diet of toothed whales, as well as other marine predators, and in pelagic food webs generally, may be underestimated.

The presence of plastic debris in the stomach of this whale, although not large or copious enough to have resulted in death, adds a further report of such debris in the stomachs of sperm whales across oceans since the 1970s (*Clarke & Young, 1998*; *Evans & Hindell, 2004*; *Clarke, Martin & Pascoe, 1993*; *Martin & Clarke, 1986*; *Roberts, 2003*; *De Stephanis et al., 2013*) and highlights the current prevalence of marine trash in the oceans. The ingestion of plastic debris has been known to result in the death of sperm whales due to gastric blockage or rupture (*De Stephanis et al., 2013*; *Jacobsen, Massey & Gull, 2010*). Further, plastic debris can also result in problems such as injury or entanglement of whales and other marine mammals (*Pace, Miragliuolo & Mussi, 2008*; *Gregory, 2009*). With the amount of marine litter (including plastic debris) generated by Southeast Asian nations equaling or exceeding global averages (*Todd, Ong & Chou, 2010*), this may be of conservation concern to threatened marine species, such as the sperm whale, in the region.

The Singapore sperm whale had a control region haplotype that is present in the northern Atlantic, northern and southwestern Pacific, central, western and southern Indian, and Southern Oceans (Haplotype 1: (*Lyrholm & Gyllensten, 1998*); Haplotype A: (*Engelhaupt et al., 2009*; *Alexander et al., 2016*)). It is among the most common haplotypes worldwide

(*Alexander et al., 2016*), and this study extends its known distribution to the Southeast Asian Indo-West Pacific. Although widespread, this haplotype appears in the highest frequency in the northern Pacific, specifically off Japanese coastal areas (*Lyrholm & Gyllensten, 1998*), and off Cocos (Keeling) Islands in the southern Indian Ocean (*Alexander et al., 2016*). In contrast, it was found to be absent off Sri Lanka (*Alexander et al., 2016*).

The coastal hydrodynamic model results suggest that the Singapore sperm whale was likely to have been to the west of Singapore prior to her being found dead off Jurong Island. Furthermore a selected group of drogue tracks as used in the model point to a likely location for the start of free drifting of the whale carcass close to the shipping routes rather than further out in the Malacca Straits.

In summary, current circumstantial evidence from the diet, origin of ingested plastic debris, mitochondrial DNA haplotype, and hydrodynamic modeling suggests that the sperm whale could have originated from a population in the Indian Ocean, close to Cocos (Keeling) Islands or Indonesia. However, until more detailed genetic sampling and kinship analyses of sperm whales off Southeast Asian waters can be done, it would not be possible to confidently determine the origin of the Singapore specimen. Although no parasites or barnacles were found on this whale, future studies of parasite presence or identity, and ageing of whales based on teeth may also help narrow down the origins of stranded animals.

This study of the diet and haplotype of a sperm whale found dead in Singapore waters represents the first opportunity to understand these aspects of sperm whale biology in the Southeast Asian Indo-West Pacific region. Although most of the dietary components were identified and described, it was not possible to determine the precise biomass contribution of each prey species. This is because (a) soft tissues of squid prey items were completely digested, (b) and mass conversion formulas are not available for all recorded species, and (c) several morphological types remain unidentified as identification guides for squid beaks from the region are not available. Also, owing to the whale's back and spinal injury, with a few of the caudal vertebrae smashed, possibly caused by a ship strike, it is assumed that the whale did not forage normally or feed for some days before her death. Hence, the true relative importance of each species in the diet of this individual may not be accurately reflected. However, the data for the majority of the species eaten, their numerical importance, and the whale's mitochondrial DNA haplotype nonetheless provides the first steps to understanding the diet of this sperm whale. Further, the combined analyses of stomach contents, DNA, and hydrodynamic modeling could provide a context to future studies on the sperm whale, and have broader applicability on other marine mammals in the region.

## CONCLUSIONS

In this study we provided an increase in the understanding the diet and natural history of the sperm whale in Southeast Asia. A dead adult female sperm whale found in Singapore fed mainly on small to medium-sized mesopelagic Indo-West Pacific squids, with a smaller proportion of other marine invertebrates and fish. The sperm whale had the most widespread and common mitochondrial DNA control region haplotype that is present in

the northern Atlantic, northern and southwestern Pacific, central, western and southern Indian, and Southern Oceans. Current circumstantial evidence from the diet, origin of ingested plastic debris, mitochondrial DNA haplotype, and hydrodynamic modeling suggests that the sperm whale could have originated from a population in the Indian Ocean. The combined analyses of stomach contents, DNA, and hydrodynamic modeling could provide a context to future studies on the sperm whale, and have broader applicability on other marine mammals in the region.

## ACKNOWLEDGEMENTS

The Singapore sperm whale salvage operation that enabled this study to be conducted was made possible by the support rendered by the dedicated staff, volunteers and contractors of the Lee Kong Chian Natural History Museum, the Maritime and Port Authority of Singapore, the National Environment Agency, and public assistance. Chen Mingshi, Foo Maosheng, Iffah Iesa, and Kate Pocklington were instrumental in the dissection of the whale. Marina Chaw, Chen Mingshi, Iffah Iesa, and Tan Siong Kiat assisted with the initial sorting of the stomach contents. Peter K.L. Ng, J.C. Mendoza, and Tohru Naruse helped with the crustacean identification. Rudolf Meier and Amrita Srivathsan provided advice on the genetic analysis. Sarah Mesnick and Thomas Lyrhomn kindly provided information and references on sperm whale genetics. We thank the peer reviewers and editors for their helpful comments.

### Funding

This study was funded by the Jubilee Whale Fund. The funders had no role in study design, data collection and analysis, decision to publish, or preparation of the manuscript.

### Grant Disclosures

The following grant information was disclosed by the authors:
Jubilee Whale Fund.

### Competing Interests

The authors declare there are no competing interests.

### Author Contributions

- Marcus A.H. Chua and Seng Keat Ooi conceived and designed the experiments, performed the experiments, analyzed the data, contributed reagents/materials/analysis tools, prepared figures and/or tables, authored or reviewed drafts of the paper, approved the final draft.
- David J.W. Lane performed the experiments, analyzed the data, authored or reviewed drafts of the paper, prepared figures and/or tables, approved the final draft.
- Serene H.X. Tay and Tsunemi Kubodera conceived and designed the experiments, performed the experiments, analyzed the data, contributed reagents/materials/analysis tools, prepared figures and/or tables, approved the final draft.

## Field Study Permissions

The following information was supplied relating to field study approvals (i.e., approving body and any reference numbers):

Permission to work on the beach was granted verbally by Boon Kiang Chew from the National Environment Agency Singapore.

## Data Availability

The raw data from the sperm whale stomach contents are available in Table S1.

## Supplemental Information

Supplemental information for this article can be found online at http://dx.doi.org/10.7717/peerj.6705#supplemental-information.

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
