# Peer review of "Diet and mitochondrial DNA haplotype of a sperm whale (Physeter macrocephalus) found dead off Jurong Island, Singapore"

_PeerJ, doi:10.7717/peerj.6705_

## Round 0.1 · original submission · Minor Revisions

I have heard back from two reviewers, both of whom are supportive of your work. They each have many constructive comments, mainly editorial in nature, that I believe you can easily address. I look forward to a re-submission.

Reviewer 1 ·

Basic reporting

Overall the basic report is quite good. However, I have some specific suggestions below.

1. Abstract - third sentence is not a complete sentence. Could change to something like “Here we report on the identification of stomach contents and mitochondrial haplotype of this individual, as well as hydrodynamic modelling used to determine the possible geographic origin of the individual.”

2. Lines 45-48: This sentences is a bit awkward. Perhaps something like “Genetic analyses have found that sperm whales are subdivided into a large number of groups based on mitochondrial DNA, whereas there is little-to-no structuring based on nuclear markers (refs). This fits with what is known about their social structure, where females show strong fidelity to specific areas, whereas males may wander around the globe.”

3. Line 93: I would change “consumed” to “present” because the animal could have consumed items without hard parts, which therefore would not be detected. I would therefore argue that it is impossible to tell everything that the animal consumed, all you can do is report on what was present in the stomach; there is a difference.

4. Line 115: Change “larger” to “large”

5. Line 126: Should be “tropical conditions”

6. Line128: Should be “ended up close to”

7. Line 214: Should be “known to forage”

Experimental design

Not so much of an issue with this paper, at least at a “big picture” scale, as it is just describing what was found in a dead sperm whale that washed ashore. However, the protocols used, and analyses conducted, are appropriate for the goals of the study and the data available.

Validity of the findings

Good. The authors summarize their findings well, and interpret them appropriately in the context of other available information on this species.

Additional comments

Overall it is a good paper. However, there is one fairly big topic that the authors should elaborate on. In lines 286-288 they mention that it had a large spinal injury, possibly caused by a ship strike. This seems extremely important, but is only given one sentence. If the authors are planning on reporting on the necropsy and its findings in a separate paper, then they should say so here (as readers will be expecting more information). Otherwise, they should greatly expand on this finding. For example, is there evidence of haemorrhaging that could indicate if the damage occurred pre- or post-mortem? Again, this is extremely important, because mortalities from ship strikes are an increasing threat to marine mammals around the world, but there are few data from this region, so this would be an important data point.

Reviewer 2 ·

Basic reporting

No comment

Experimental design

No comment

Validity of the findings

No comment

Additional comments

I have previously seen this manuscript when it was submitted to PLOS ONE (don’t worry, I was the supportive Reviewer #1!). I have checked to see whether previous comments on the manuscript were addressed (most of which were), and if not, have re-mentioned them here (so I can see in your reply letter why you chose not to include those suggestions – perfectly fine if there is a valid reason...which might just be hoping not to get the same reviewer again!), as well as doing a fresh read through of the manuscript. I stand by my statement on the last version of the manuscript – this paper does a great job gleaning as much as possible from a stranding event in order to surmise what is going on in the nearby sperm whale population, and I’m very glad the authors have submitted it to a different journal. I have just minor editorial comments below.

Abstract
Line 24: Might be worth pointing out that one of the reasons that mtDNA haplotypes are worth looking at in this species are the strongly defined patterns of female philopatry.
Line 28: The comma following ‘Mesopelagic squids’, makes it seem like mesopelagic squids and then these specific species were found in the sperm whale. If these specific species are mesopelagic squids then it would be better to say ‘The mesopelagic squids X, Y, Z’ and then list the species names.
Lines 33-34: This is a pretty audacious claim - there have been other studies on sperm whales in south-eastern Asia so I think you need to modify the language to make it sound less like nothing is known of sperm whales from this area.
Line 34-36: Suggest modifying the language of this sentence to “The combined analyses of stomach contents, DNA, and hydrodynamic modelling could provide a context for future studies of sperm whale strandings, and have broader applicability for other marine mammals in the region.”

Introduction:
Line 40-41: “They are typically found in regions with deep seas” seems a little stilted. Potentially, “They are typically found in deep, off-shore waters”.
Lines 45-48: This sentence seems a bit overcomplicated. What about something along the lines of: “Sperm whale population structure and studies of the maternally-inherited mitochondrial DNA have shown differentiation between populations and suggest female philopatry, which has also been of interest to biologists [13–17].”
Line 51: “western” should not be capitalised – description, not location.
Line 65: Suggest clarifying why the fully erupted teeth point is important (e.g. ageing etc)
Lines 72-73: Suggest focusing on the broader take-homes I mentioned in response to comments on the abstract rather than on this narrower take.

Methods:
Line 77: Was her age-class as an adult established because of her size? This would be a good point to add in if so.
Line 85: Think this reference should be pointing to [28] not [27]
Lines 91: Could these references just read [27-31]?
Line 93: By number, or by estimated biomass?
Line 99: “sequences” rather than “sequencings”
Line 98-99: How many replicates? Two replicates for each tissue type? Need to be more specific.
Line 114: Think you can ditch the “the” in front of “larger tropical storm”
Line 116: Suggest removing ‘model called the”, adding “was” before built, and “and” before was used

Results:
Line 137: Again, need to clarify whether percentage total is by number or biomass. Also suggest removing the word “represented”.
Line 138: Need to point out that this estimated mass is per individual squid, not in aggregate across all Taonius pavo.
Line 143: “was” not “is”
Line 165: “included” not “include”
Line 171: “Mitochondrial DNA haplotype identification” instead of “haplotype identification”

Discussion:
Although the authors do a good job summarising the broader implications of the diet analysis to lead off the discussion, I think you might need a broader opening highlighting the other main findings of your study
Line 192: But were fish were underrepresented due to digestion pressure?
Line 197: Need a space between typically and have.
Line 198: “different the different” – remove the first different.
Line 210: You can remove “a finding” from this sentence.
Line 211: Add an “a” before majority.
Line 214: Add a “to” before forage.
Line 223: Suggest changing “their” to “the male’s” because there is some confusion about what the subject of this sentence is e.g. the pyrosomes or the male sperm whales.
Line 241-242: Interesting point!
Line 258: Think this should be “common haplotypes” not “common haplotype”
Line 286-288: Or could the spinal injury have caused her immediate death?

Conclusion:
Line 297-298: Again, same comment about broadening this sentence out to make it seem less like it is the first research on sperm whales in southeastern Asia.
Line 300: Add “mitochondrial DNA” before “control”

Tables and Figures:
Table 1: I feel like this table might be better presented as a series of histograms/bar graphs/scatterplots representing this information graphically rather than in a long-form table. Would make it easier to compare between species at least. Either way, will need to clarify for the figure legend whether this percentage total is calculated by number or estimated biomass.

Figure 1 legend: suggest adding “simulated” in front of drogues

Figure 4: Add location of whale when it was located. Suggest adding “simulated” in front of first mention of “the drogues”. Line 187: suggest changing wording “that will end up” to “to have ended up”. Also suggest adding “on 10 July 2015” (because of the point you made in the manuscript that there are different strong seasonal effects).

Supplementary Materials
The video is a cool resource – I look forward to the day when such materials can actually be embedded in articles. One modification if it is not too difficult to make (and not necessary if it is), would be to mark the location of where the whale stranded on Jurong Island on this animation, so folks can easily see which of the drift simulations could have resulted in the whale being deposited there (similar to recommendations for Fig 4, and what the authors have already done for Fig 1).

The table with the squid lengths in it needs a little more context (potentially a readme tab?) so that it can be interpreted if someone just opens the file e.g. define LRL, what are the units, what is “-“ given for some records (e.g. beak too degraded to get measurements off?)?

---

## Round 0.2 · accepted · Accept

I have gone over your revisions and responses to reviewers, and this paper is well revised. Thus, I am happy to move this into production, and I look forward to seeing the published version!

#